# Flotillin-mediated membrane fluidity controls peptidoglycan synthesis and MreB movement

Aleksandra Zielińska[1†], Abigail Savietto[2,3†], Anabela de Sousa Borges[1], Denis Martinez[4], Melanie Berbon[4], Joël R Roelofsen[1], Alwin M Hartman[5,6,7], Rinse de Boer[8], Ida J Van der Klei[8], Anna KH Hirsch[5,6,7], Birgit Habenstein[4], Marc Bramkamp[2,3]*, Dirk-Jan Scheffers[1]*

[1]Molecular Microbiology, Groningen Biomolecular Sciences and Biotechnology Institute, University of Groningen, Groningen, Netherlands; [2]Biozentrum, Ludwig-Maximilians-Universität München, München, Germany; [3]Institute for General Microbiology, Christian-Albrechts-University, Kiel, Germany; [4]Institute of Chemistry & Biology of Membranes & Nanoobjects (UMR5248 CBMN), IECB, CNRS, Université Bordeaux, Institut Polytechnique Bordeaux, Pessac, France; [5]Department of Drug Design and Optimization (DDOP), Helmholtz-Institute for Pharmaceutical Research Saarland (HIPS) - Helmholtz Centre for Infection Research (HZI), Saarbrücken, Germany; [6]Department of Pharmacy, Saarland University, Saarbrücken, Germany; [7]Stratingh Institute for Chemistry, University of Groningen, Groningen, Netherlands; [8]Molecular Cell Biology, Groningen Biomolecular Sciences and Biotechnology Institute, University of Groningen, Groningen, Netherlands

*For correspondence:
bramkamp@ifam.uni-kiel.de (MB);
d.j.scheffers@rug.nl (D-JS)

[†]These authors contributed equally to this work

Competing interests: The authors declare that no competing interests exist.

**Abstract** The bacterial plasma membrane is an important cellular compartment. In recent years it has become obvious that protein complexes and lipids are not uniformly distributed within membranes. Current hypotheses suggest that flotillin proteins are required for the formation of complexes of membrane proteins including cell-wall synthetic proteins. We show here that bacterial flotillins are important factors for membrane fluidity homeostasis. Loss of flotillins leads to a decrease in membrane fluidity that in turn leads to alterations in MreB dynamics and, as a consequence, in peptidoglycan synthesis. These alterations are reverted when membrane fluidity is restored by a chemical fluidizer. In vitro, the addition of a flotillin increases membrane fluidity of liposomes. Our data support a model in which flotillins are required for direct control of membrane fluidity rather than for the formation of protein complexes via direct protein-protein interactions.

## Introduction

The shape of a bacterium is predominantly defined by the structure of its peptidoglycan. Although there is a great variety in bacterial shapes, the overall chemistry of peptidoglycan is very similar between bacteria and thus the shape of peptidoglycan is primarily determined by the temporal and spatial regulation of peptidoglycan synthesis. In rod-shaped bacteria, peptidoglycan synthesis is thought to be mediated by two protein assemblies, the elongasome and the divisome, that synthesise peptidoglycan along the long axis and across the division plane of the cell, respectively (*Typas et al., 2012*; *Zhao et al., 2017*). These complexes contain a set of proteins required for the final steps of synthesis and translocation of the peptidoglycan precursor, LipidII, from the inner to the outer leaflet of the cytoplasmic membrane, and proteins that incorporate LipidII into peptidoglycan. These include SEDS (Shape, Elongation, Division and Sporulation) proteins that can perform

**eLife digest** Every living cell is enclosed by a flexible membrane made of molecules known as phospholipids, which protects the cell from harmful chemicals and other threats. In bacteria and some other organisms, a rigid structure known as the cell wall sits just outside of the membrane and determines the cell's shape.

There are several proteins in the membrane of bacteria that allow the cell to grow by assembling new pieces of the cell wall. To ensure these proteins expand the cell wall at the right locations, another protein known as MreB moves and organizes them to the appropriate place in the membrane and controls their activity. Previous studies have found that another class of proteins called flotillins are involved in arranging proteins and phospholipid molecules within membranes. Bacteria lacking these proteins do not grow properly and are unable to maintain their normal shape. However, the precise role of the flotillins remained unclear.

Here, Zielińska, Savietto et al. used microscopy approaches to study flotillins in a bacterium known as *Bacillus subtilis.* The experiments found that, in the presence of flotillins, MreB moved around the membrane more quickly (suggesting it was more active) than when no flotillins were present. Similar results were observed when bacterial cells lacking flotillins were treated with a chemical that made membranes more 'fluid' – that is, made it easier for the molecules within the membrane to travel around. Further experiments found that flotillins allowed the phospholipid molecules within an artificial membrane to move around more freely, which increases the fluidity of the membrane.

These findings suggest that flotillins make the membranes of bacterial cells more fluid to help cells expand their walls and perform several other processes. Understanding how bacteria control the components of their membranes will further our understanding of how many currently available antibiotics work and may potentially lead to the design of new antibiotics in the future.

glycosyl transferase reactions (*Cho et al., 2016*; *Meeske et al., 2016*; *Taguchi et al., 2019*), and Penicillin Binding Proteins (PBPs) that are divided in class A PBPs (aPBPs) that catalyse both glycosyl transferase and transpeptidase reactions, class B PBPs (bPBPs) that only catalyse transpeptidase reactions and low molecular weight PBPs that modify peptidoglycan, as well as hydrolases (*Zhao et al., 2017*; *Morales Angeles and Scheffers, 2017*).

Coordination of these complexes is linked to cytoskeletal elements, MreB (-like proteins) for the elongasome and FtsZ for the divisome. In models, the cytoplasmic membrane is often depicted as a passive environment in which these machineries are embedded. However, it is becoming clear that the structure of the membrane plays a critical role in the coordination of peptidoglycan synthesis (*Strahl and Errington, 2017*). Inward membrane curvature serves as a localisation trigger for MreB and the elongasome, and enhanced local synthesis at bulges straightens out the membrane sufficient to convert spherical cells to a rod shape (*Hussain et al., 2018*; *Ursell et al., 2014*). In *Bacillus subtilis,* the motion of MreB along the membrane is associated with elongasome activity (*Domínguez-Escobar et al., 2011*; *Garner et al., 2011*), and the velocity of MreB patches is related to growth rate (*Billaudeau et al., 2017*), indicating that MreB motion can be used as a marker for elongasome activity. Interestingly, MreB localises to and organises regions of increased membrane fluidity (RIF) (*Strahl et al., 2014*), which in turn is linked to the presence of LipidII, which favours a more fluid membrane and promotes local membrane disorder (*Ganchev et al., 2006*; *Witzke et al., 2016*). Inhibition of LipidII synthesis by genetic or chemical means results in a dissolution of membrane structures observed with the dye FM 4–64 and release of MreB from the membrane (*Domínguez-Escobar et al., 2011*; *Garner et al., 2011*; *Muchová et al., 2011*; *Schirner et al., 2015*).

Next to RIFs, membrane regions of decreased fluidity have been identified in bacteria (*Strahl and Errington, 2017*; *Bramkamp and Lopez, 2015*; *Lopez and Koch, 2017*). These so-called functional membrane microdomains (FMMs) are thought to be organised by the bacterial flotillin proteins, are enriched in isoprenoid lipids (*García-Fernández et al., 2017*; *López and Kolter, 2010*), and can be found in so-called Detergent Resistant Membrane (DRM) fractions of the membrane. Since the formulation of the FMM hypothesis, FMMs have been linked to many processes, such as protein secretion, biofilm formation, competence and cell morphology (*Mielich-Süss and Lopez, 2015*; *Mielich-*

*Süss et al., 2013*; *Bach and Bramkamp, 2013*; *Dempwolff et al., 2012*). Cell morphology defects are linked to cell wall synthesis, and analysis of the protein content of *Bacillus subtilis* DRMs identified several PBPs, MreC and other proteins involved in cell wall metabolism as well as the two flotillins, FloA and FloT (*López and Kolter, 2010*; *Bach and Bramkamp, 2013*; *Yepes et al., 2012*). FloA is constitutively expressed, whereas FloT is expressed primarily during stationary growth, cell wall stress and sporulation (*Schneider et al., 2015a*; *Huang et al., 1999*; *Nicolas et al., 2012*). Super resolution microscopy showed that the flotillins and other proteins found in DRMs do not colocalise and have different dynamics (*Dempwolff et al., 2016*), so it is unlikely that FMMs are regions in the membrane that offer a favourable environment in which these membrane proteins are continuously present and active. Recently, the hypothesis has been put forward that FMMs/flotillins form a platform for the formation of functional protein oligomers, as work in *Staphylococcus aureus* showed that multimerisation of Type 7 secretion systems and PBP2a depends on FMMs (*Lopez and Koch, 2017*; *García-Fernández et al., 2017*; *Mielich-Süss et al., 2017*).

Here, we have analysed the role of flotillins in peptidoglycan synthesis in *B. subtilis.* Our results show that, at high growth rates, flotillins control membrane fluidity in a manner that is critical for peptidoglycan synthesis and MreB dynamics, but have no effect on PBP oligomerisation. This results in a new model for flotillin function in the physical organisation of membranes during fast growth.

## Results

### Absence of flotillins shifts peptidoglycan synthesis to division Septa

In previous studies, a double deletion of *floA/floT* was either reported to suffer severe shape defects and perturbed membrane structure (*Dempwolff et al., 2012*), or to not have strong shape defects but with a change in the overall lipid ordering of the membrane (*Bach and Bramkamp, 2013*). We grew wild type and Δ*floAT* strains and analysed exponentially growing cells. We did not observe striking shape defects but did see an increase in median cell length and distribution of cell lengths in the absence of flotillins (*Figure 1A,G*). To look at effects on peptidoglycan synthesis, we labelled cells with HADA, a fluorescent D-Alanine analogue that reports on sites of active peptidoglycan synthesis (*Kuru et al., 2012*), and with fluorescent vancomycin (Van-FL), which labels LipidII and peptidoglycan containing pentapeptide side chains (*Daniel and Errington, 2003*; *Morales Angeles et al., 2017*). This revealed a significant accumulation of peptidoglycan synthesis stains at division septa in the Δ*floAT* strain (*Figure 1A–C*). To look at membrane structure, cells were labelled with FM4-64, Nile-Red and DiI-C12, which are lipid dyes that accumulate in zones enriched in fluid lipids (*Strahl et al., 2014*). Again, the stains accumulated at the septa in the Δ*floAT* strain, which also showed some accumulation of FM4-64 and DiI-C12 in patches, suggesting that the more fluid regions of the membrane are coalescing into larger regions (*Figure 1A,D–F*). The HADA, FM4-64 and Nile-Red measurements were repeated using a wild type strain expressing endogenous GFP, allowing simultaneous imaging of both strains on the same slide, and gave similar results, confirming that the observed signal increase is not due to variation between microcopy experiments (*Figure 1— figure supplement 1A,B*). In this mixed-strain experiment, Nile-Red labelling at the lateral membrane was the same between wild type and Δ*floAT* strains, indicating that there is no difference in dye diffusion between the strains (*Figure 1—figure supplement 1D*). Inspection of the septa by electron microscopy revealed that there was no difference between the thicknesses of the septa between the wild type and Δ*floAT* strain, ruling out that the increase in signal was due to formation of thicker septa (*Figure 1—figure supplement 1C*). The shift of peptidoglycan synthesis to the division site could hint at stress in the overall peptidoglycan synthesis route. This was confirmed by growing cells at a sublethal concentration of fosfomycin, which limits synthesis of LipidII (*Kahan et al., 1974*), but that does not impact growth rate at the concentration used. This resulted in bulging cells and some lysis, which was exacerbated in the Δ*floAT* strain (*Figure 1—figure supplement 1F,G*). It should be noted that the peptidoglycan synthesis stress caused by fosfomycin is not the same as the stress caused by the absence of flotillins, as the phenotypes of wild type cells with sublethal fosfomycin are quite distinct from Δ*floAT* cells without fosfomycin.

We ruled out that the peptidoglycan synthesis stress was caused by a change in the folding or complex formation by PBPs in the absence of flotillins, as there were no differences in the overall PBP-profiles of Bocillin-FL labelled wild type or flotillin deletion strains (*Figure 1—figure*

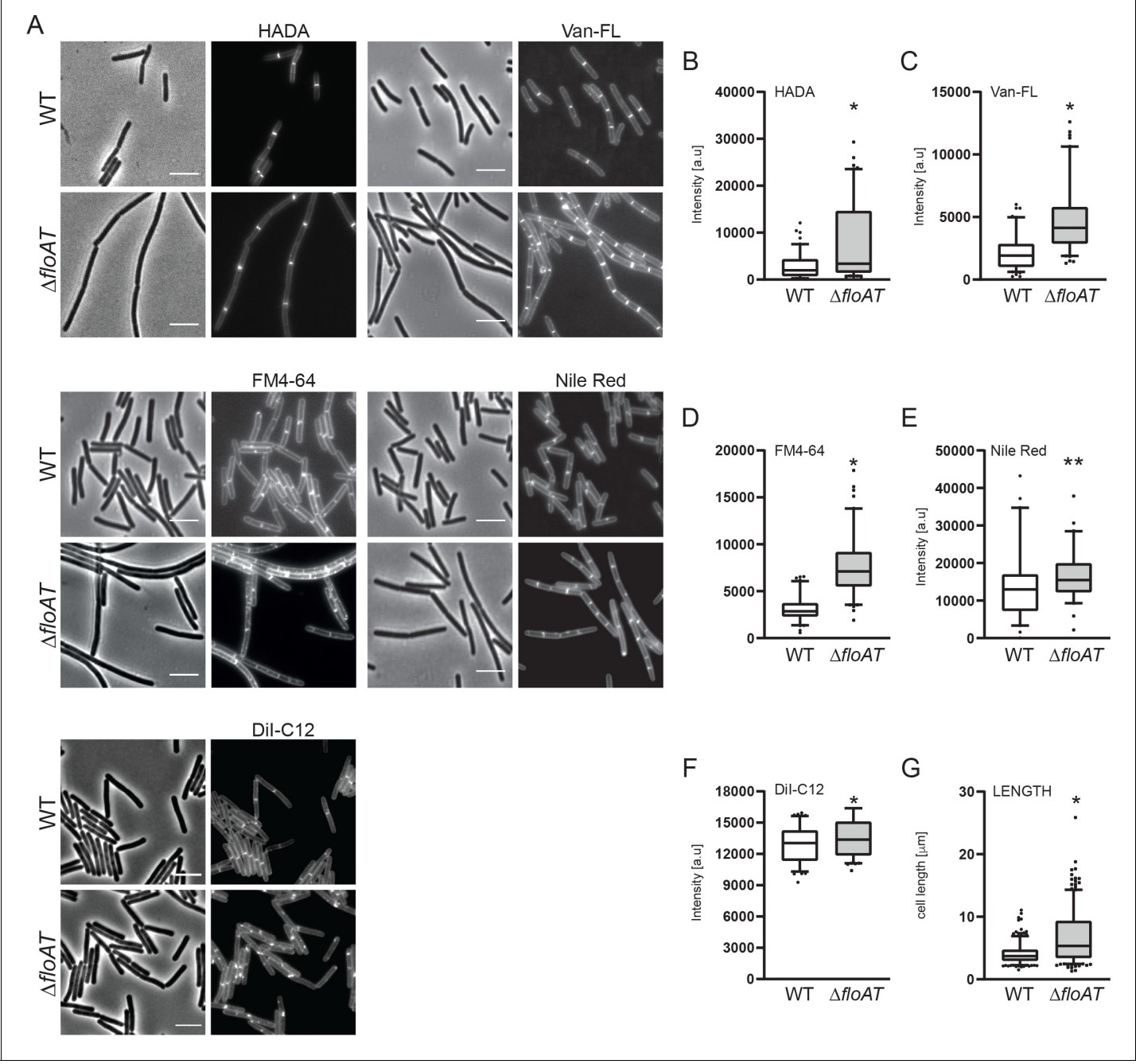

**Figure 1.** Accumulation of peptidoglycan synthesis and membrane material at division sites in a flotillin mutant. (A) Morphology of the exponentially growing wild type (WT) and Δ*floAT* strains labelled with HADA, fluorescent Vancomycin (Van-FL), FM 4–64, Nile Red, and DiI-C12. Scale bar: 5 μm. (B–F) Peak intensity of HADA (B), Van-FL (C), Nile Red (D), FM4-64 (E) and DiI-C12 (F) labelled division sites of the cells shown in (A). Cells from each strain (n ≥ 100, except E, n = 60) were analysed using the ObjectJ macro tool PeakFinder followed by statistical analysis with Prism. Significant differences are based on the two-tailed Mann-Whitney test (*p<0.05; **p<0.01). (G) Distribution of the cell length of the strains analysed in (A). Statistical analysis of the data (n = 100, two tailed Mann-Whitney test, *p<0.05) was performed with Prism, resulting in box plot graphs.

The online version of this article includes the following source data and figure supplement(s) for figure 1:

**Source data 1.** Fluorescence intensity and cell length measurements.

**Figure supplement 1.** Control experiments showing that differences in septal labeling intensity are not due to microscopy settings, septum thickness, or dye diffusion.

**Figure supplement 1—source data 1.** Growth data plotted in FS1F.

**Figure supplement 1—source data 2.** Data plotted in F1F1Sb, F1FS1C, F1FS1.

*Figure 1 continued on next page*

*Figure 1 continued*

**Figure supplement 2.** Absence of flotillins does not affect expression, oligomerisation or localisation of PBPs.

*supplement 2A*). PBP complex formation was analysed using a combination of Native-PAGE and SDS-PAGE with Bocillin-labelled membrane fractions (*Trip and Scheffers, 2016*) and showed that various PBPs can be found in a high-MW complex (notably PBPs 1, 2, 3 and 4), but that complex formation is similar in the ΔfloAT strain (*Figure 1—figure supplement 2B*). Also, none of the five functional GFP-PBPs examined changed their localisation in the ΔfloAT strain (*Figure 1—figure supplement 2C*). Overall, the data suggest that in the absence of flotillins, peptidoglycan synthesis is affected and relatively increased at division septa, with a concomitant accumulation of membrane dyes that are indicative of higher membrane fluidity.

## The absence of both flotillins and PBP1 causes a severe phenotype, linked to a loss of membrane fluidity

We reasoned that a non-lethal defect in septal peptidoglycan synthesis could reveal more about the role of flotillins and constructed a flotillin mutant that lacks PBP1, a bifunctional glycosyl transferase/transpeptidase that is required for efficient cell division (*Scheffers and Errington, 2004*). Simultaneous deletion of *pbp1*, *floA*, and *floT* resulted in strong filamentation and delocalisation of peptidoglycan synthesis as well as membrane dyes to patches (*Figure 2A*, *Figure 2—figure supplement 1A*). Deletion of single flotillin genes and PBP1 had similar, albeit less severe effects (*Figure 2—figure supplement 1B,C*). To exclude the possibility that an alteration of peptidoglycan modification resulted in the delocalisation of HADA and Van-FL, we used D-Alanine-D-Propargylglycine (D-Ala-D-Pra), a clickable dipeptide analogue which is exclusively incorporated into peptidoglycan via LipidII (*Sarkar et al., 2016*). D-Ala-D-Pra incorporation was delocalised in the Δpbp1ΔfloAT strain, indicating that peptidoglycan synthesis itself is delocalised (*Figure 2—figure supplement 1D*). So far, our experiments were done with fast growing cells and Lysogeny Broth (LB) as the growth medium. Strikingly, none of the mutant strains had an apparent phenotype when cultivated in Spizizen's minimal medium (SMM, *Figure 2B*), and peptidoglycan synthesis and lipid dyes were no longer accumulating at division sites in the ΔfloAT strain (*Figure 2—figure supplement 2*). SMM has a higher $Mg^{2+}$ concentration, which is known to rescue various cell shape mutations by inhibition of cell wall hydrolysis (*Dajkovic et al., 2017*). However, the increase in $Mg^{2+}$ was not sufficient to explain the reversal of phenotype as cells grown on LB supplemented with $Mg^{2+}$ (6 mM, concentration in SMM, or 20 mM) still displayed the elongated phenotype with delocalised peptidoglycan synthesis (*Figure 2—figure supplement 3*). This indicated that the phenotypes associated with the absence of flotillins are growth-rate and/or nutrient related.

Next, we determined lipid packing order in the different strains using the fluorescent dye Laurdan, a reporter for flotillin-mediated lipid ordering (*Bach and Bramkamp, 2013*). LB-grown cells lacking flotillins displayed an increased generalised polarisation (GP) (*Bach and Bramkamp, 2013*), indicative of an overall increase in ordered lipid packing in the membrane, but the effect of flotillins on membrane ordering completely disappeared when cells were grown on SMM (*Figure 3*). The resolution obtained with Laurdan does not allow the detection of local differences in fluidity between the lateral membrane and the septa, but does report on overall lipid ordering. Overall, lipid order was increased in cells grown on SMM compared to LB (*Figure 3*), whereas the absence of PBP1 had no significant effect on membrane fluidity, also not when combined with flotillin deletions (*Figure 3*). The changes in lipid ordering were not due to changes in the overall fatty acid composition of the membranes - the ratios of C17/C15 side chains and *iso/anteiso* fatty acids, which are indicative of fluidity (*Strahl et al., 2014*), were identical for wild type and ΔfloAT strains grown on LB, and very similar for cells grown on SMM (*Figure 3—figure supplement 1*).

## Restoring membrane fluidity rescues normal peptidoglycan synthesis

The GP values indicated that membranes are more ordered when cells are grown on minimal medium, and this suggests that the flotillin-associated increase in overall membrane fluidity is important for cell shape control at high growth rates. This was tested by growing the strains lacking flotillins and PBP1 on LB in the presence of benzyl alcohol, an extensively used membrane fluidiser that

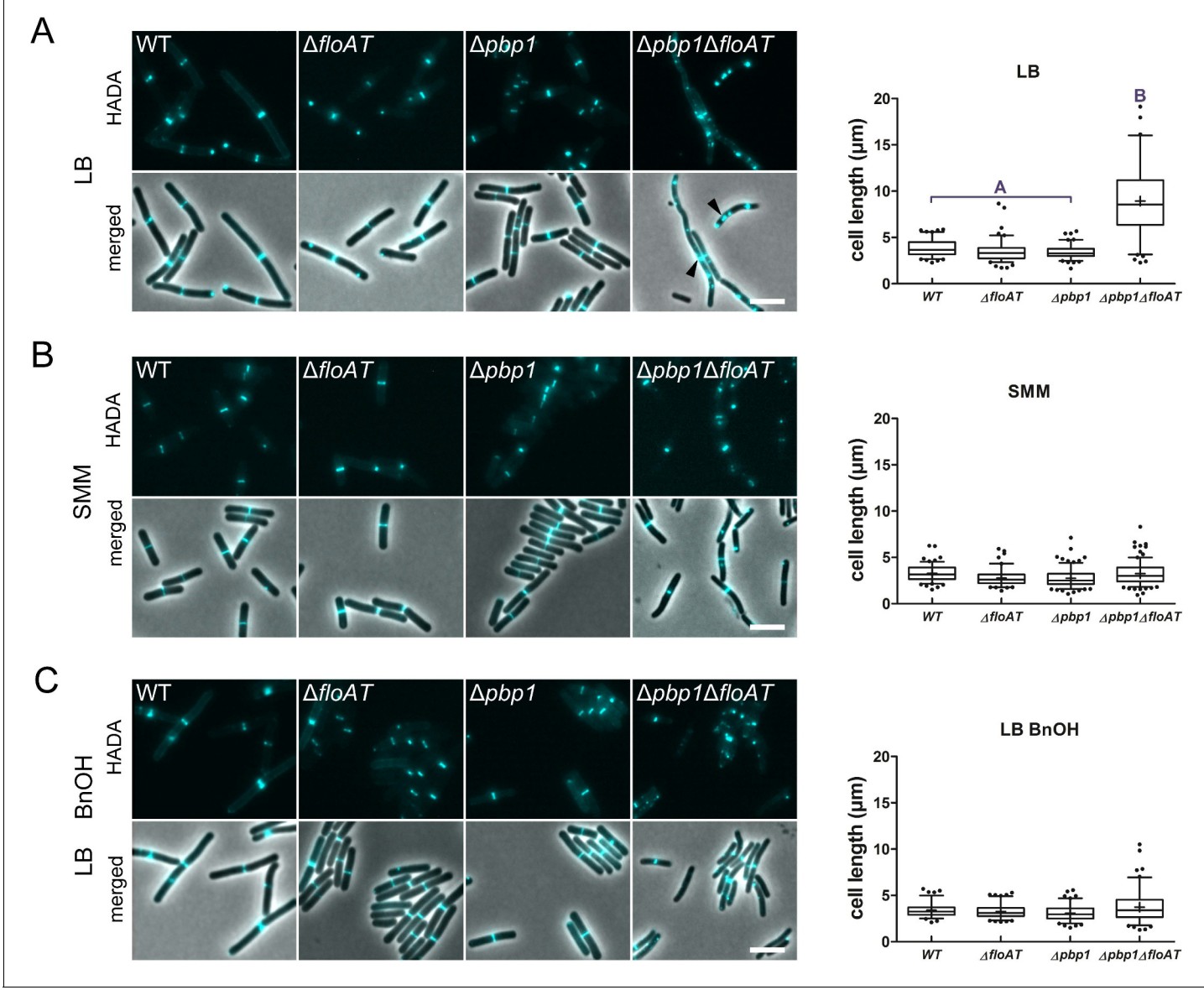

**Figure 2.** Cell morphology and cell wall synthesis localisation is dependent on growth conditions. Morphology of the *WT*, *ΔfloAT*, *Δpbp1*, and *Δpbp1ΔfloAT* strains grown in (**A**) rich (LB), (**B**) minimal (SMM) medium, and in (**C**) rich medium with membrane fluidising conditions (0.1% benzyl alcohol, LB+BnOH). Cells were labelled with HADA, and aberrant cell shape and peptidoglycan synthesis are indicated with arrowheads. Panels on the right indicate corresponding cell length distributions (n ≥ 100). Distributions were analysed using Dunn's multiple comparison tests after Kruskal–Wallis. Statistically significant cell length distribution classes (p<*0.001*) are represented as letters above each graph – in B and C there were no significant differences. Scale bar: 4 μm.

The online version of this article includes the following source data and figure supplement(s) for figure 2:

**Source data 1.** Cell length measurements.
**Figure supplement 1.** Deletion of both flotillins and PBP1 induces filamentation and delocalisation of peptidoglycan synthesis.
**Figure supplement 1—source data 1.** Cell length measurements plotted in F2F1C.
**Figure supplement 2.** Septum labelling of wild type and flotillin mutant cells grown on minimal medium.
**Figure supplement 2—source data 1.** Fluorescence intensity measurements plotted in F2FS2.
**Figure supplement 3.** Filamentation and delocalisation of peptidoglycan synthesis in the absence of flotillins and PBP1 is not rescued by the addition of magnesium.
**Figure supplement 4.** Growth curves and growth rates show similar growth for wt, *ΔfloAT*, *Δpbp1*, and *Δpbp1ΔfloAT* (as well as *ΔfloA*, *ΔfloT*, *Δpbp1ΔfloA and Δpbp1ΔfloT)* strains grown on LB or on LB supplemented with BnOH (0.1% (w/v)).
**Figure supplement 4—source data 1.** Growth curve data.

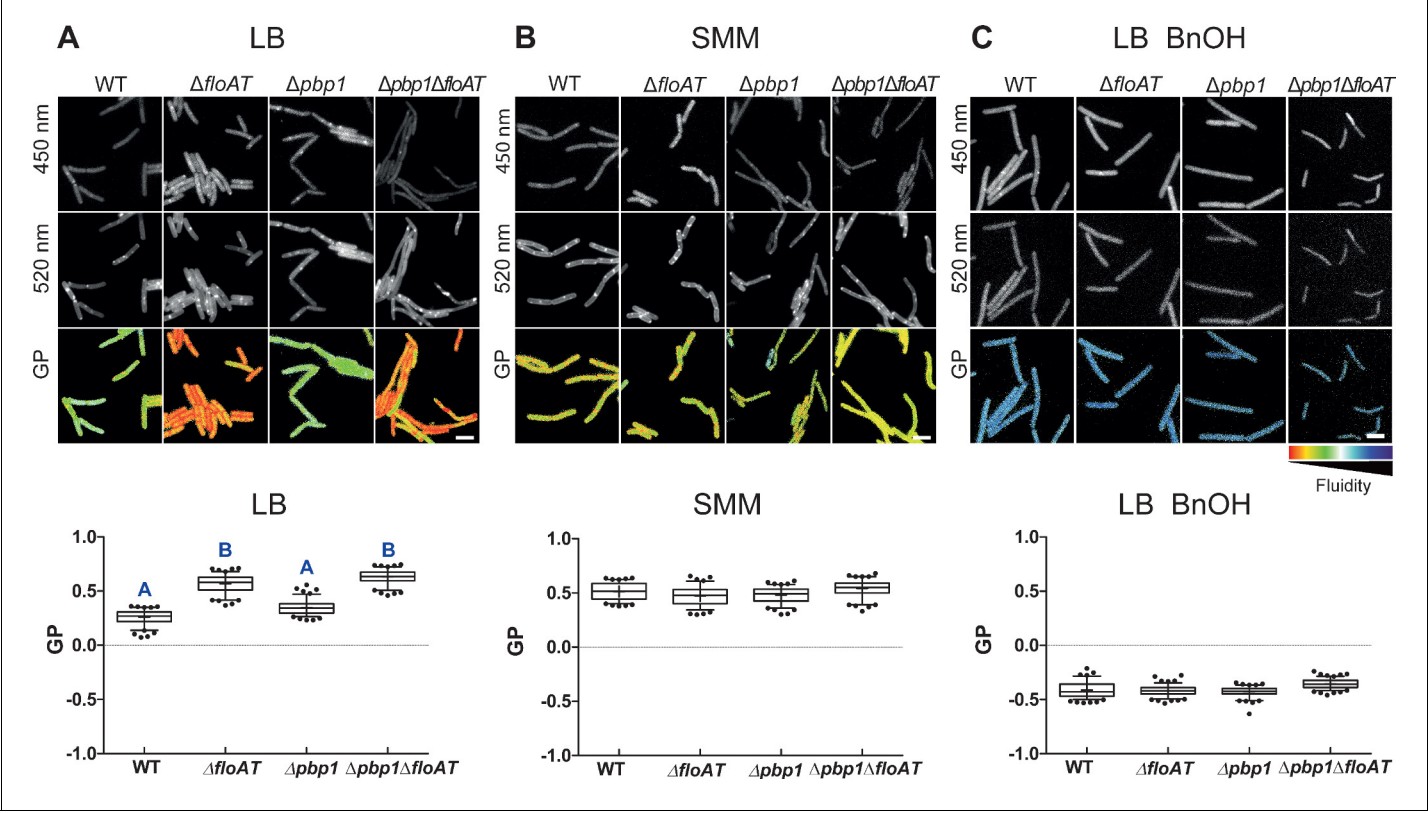

**Figure 3.** Flotillins increase overall membrane fluidity at high growth rate. Changes in overall membrane fluidity were assessed by Laurdan microscopy in cells grown on LB (A), SMM (B) and LB+BnOH (C). Micrographs show colour-coded generalised polarisation (GP) maps in which red indicates regions of decreased fluidity (scale bar: 4 μm). Correspondent theoretical GP measurements in the graphs vary from −1 (more fluid) to 1 (less fluid). Significant statistical differences according to Dunn's multiple comparison tests after Kruskal–Wallis are represented as letters above each graph in panel (A). Data labelled 'A' are significantly different from data labelled 'B'; data with the same letter are not significantly different. No statistically significant difference was observed for the data in panels (B) and (C) (p<0.001; n ≥ 150, two biological replicates).

The online version of this article includes the following source data and figure supplement(s) for figure 3:

**Source data 1.** GP measurement.
**Figure supplement 1.** Fatty acid composition analysis.
**Figure supplement 1—source data 1.** Fatty acid composition data.

increases membrane hydration due to disordering of membrane structure (*Konopásek et al., 2000*). Notably, the addition of benzyl alcohol increased membrane fluidity to similar extents in the wild-type and the mutant strains (see *Figure 3C*), but did not affect the growth rates of the strains (*Figure 2—figure supplement 4*). The increase in membrane fluidity restored normal cell length and normal peptidoglycan synthesis patterns to the *pbp1/floA/floT* strain (*Figure 2C*).

In *B. subtilis*, the rate of growth and of peptidoglycan synthesis is linked to the speed of MreB movement – in minimal media, the speed of MreB patches is reduced compared to the speed in rich media (*Billaudeau et al., 2017*). Analysis of the movement of a fully functional mRFPruby-MreB fusion (*Domínguez-Escobar et al., 2011*) by time lapse TIRF (Total Internal Reflection Fluorescence) microscopy, confirmed that MreB patch mobility is higher in cells grown on LB than in cells grown on SMM, with MreB speeds similar to those reported previously (*Billaudeau et al., 2017*; *Figure 4*, *Figure 4—videos 1* and *2*). Strikingly, in the absence of flotillins, MreB patch mobility was notably decreased in cells grown on LB, while in SMM grown cells MreB patch mobility was independent of the presence of flotillins (*Figure 4*, *Figure 4—videos 3* and *4*). Fluidising the membrane with benzyl alcohol, which does not alter the growth rate, almost completely restored MreB mobility in LB grown cells (*Figure 4*, *Figure 4—videos 5* and *6*). These results indicate that the MreB patch mobility is not only controlled by growth rate, but also by membrane fluidity. Thus, in fast growing cells with decreased membrane fluidity there is a decrease in elongasome mediated peptidoglycan synthesis,

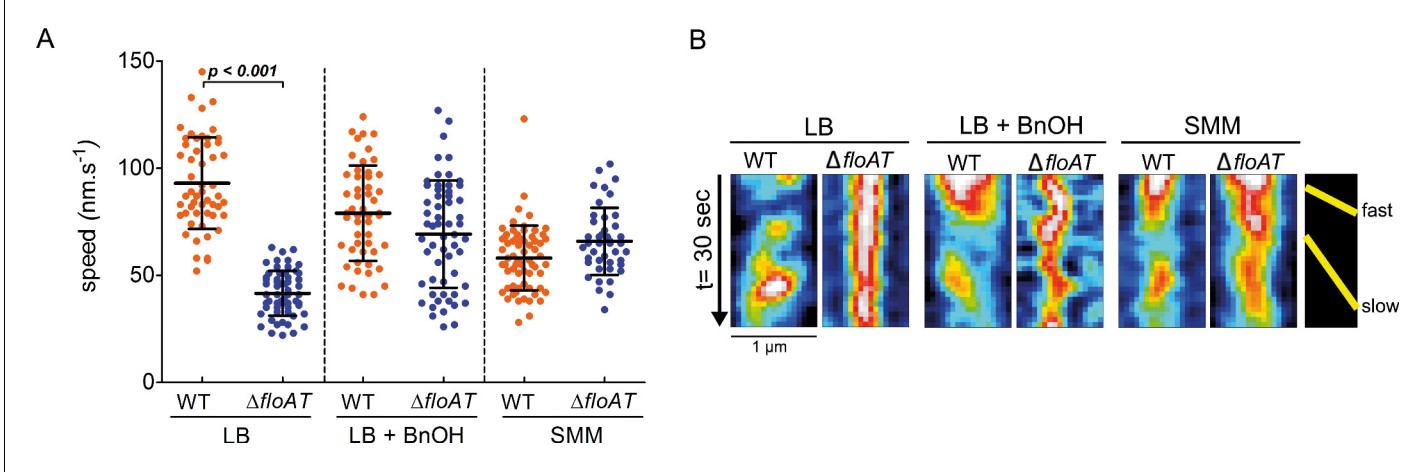

**Figure 4.** MreB speed is linked to membrane fluidity. (**A**) The MreB speed in different strain backgrounds and growth conditions was analysed by time-lapse TIRF microscopy. Scatter plot of the speed of patches obtained from individual tracks in 5 different cells are represented per fusion and condition. Average speeds are shown; error bars indicate the standard deviation. Significant statistical differences according to Dunn's multiple comparison tests after Kruskal–Wallis are represented ($p<0.001$). (**B**) Representative kymographs showing fast and slow moving patches of mRFPruby-MreB in *B. subtilis* cells lacking endogenous *mreB* (WT) or *mreB* and *floAT* (Δ*floAT*). See *Figure 4—videos 1–6* for corresponding raw image series. The online version of this article includes the following video and source data for figure 4:

**Source data 1.** MreB patch mobility measurements determined by TIRFM.

**Figure 4—video 1.** Visualisation of xylose inducible mrfpRuby-MreB patches dynamics (strain 4070) during exponential growth in LB medium at 37°C by TIRF microscopy.

https://elifesciences.org/articles/57179#fig4video1

**Figure 4—video 2.** Visualisation of xylose inducible mrfpRuby-MreB patches dynamics (strain 4070) during exponential growth in SMM medium at 37°C by TIRF microscopy.

https://elifesciences.org/articles/57179#fig4video2

**Figure 4—video 3.** Visualisation of xylose inducible mrfpRuby-MreB in Δ*floAT* patches dynamics (strain 4076) during exponential growth in LB medium at 37°C by TIRF microscopy.

https://elifesciences.org/articles/57179#fig4video3

**Figure 4—video 4.** Visualisation of xylose inducible mrfpRuby-MreB in Δ*floAT* patches dynamics (strain 4076) during exponential growth in SMM medium at 37°C by TIRF microscopy.

https://elifesciences.org/articles/57179#fig4video4

**Figure 4—video 5.** Visualisation of xylose inducible mrfpRuby-MreB patches dynamics (strain 4070) during exponential growth in LB medium supplemented with BnOH (0.1%) at 37°C by TIRF microscopy.

https://elifesciences.org/articles/57179#fig4video5

**Figure 4—video 6.** Visualisation of xylose inducible mrfpRuby-MreB in Δ*floAT* patches dynamics (strain 4076) during exponential growth in LB medium supplemented with BnOH (0.1%) at 37°C by TIRF microscopy.

https://elifesciences.org/articles/57179#fig4video6

reflected by the reduction of MreB mobility. This fits with an observed increase in peptidoglycan synthesis at the division site which may act as a compensatory mechanism.

## Flotillin increases fluidity of model membranes in vitro

To assess whether the influence of flotillins on membrane fluidity is direct, we determined the membrane fluidity of model membranes with purified flotillin using solid-state NMR (ssNMR). $^2$H ssNMR is a biophysical tool that assesses lipid mobility in native-like model membranes on the atomic level, by monitoring the carbon-deuterium order parameter of a deuterated lipid along the acyl chain (here POPC-d31) (*Molugu et al., 2017*; *Legrand et al., 2019*). We purified *B. subtilis* FloT and tested the impact of FloT on the membrane, when reconstituted in POPC-d31 liposomes (Schematically depicted in *Figure 5A*). FloT decreases the spectral width of the $^2$H quadrupolar splitting, reflecting an increase in motion on the atomic scale (*Figure 5B*). The $^2$H spectrum encodes the local order parameter $S_{CD}$ of the carbon-deuterium in absence and in presence of FloT. Strikingly, FloT

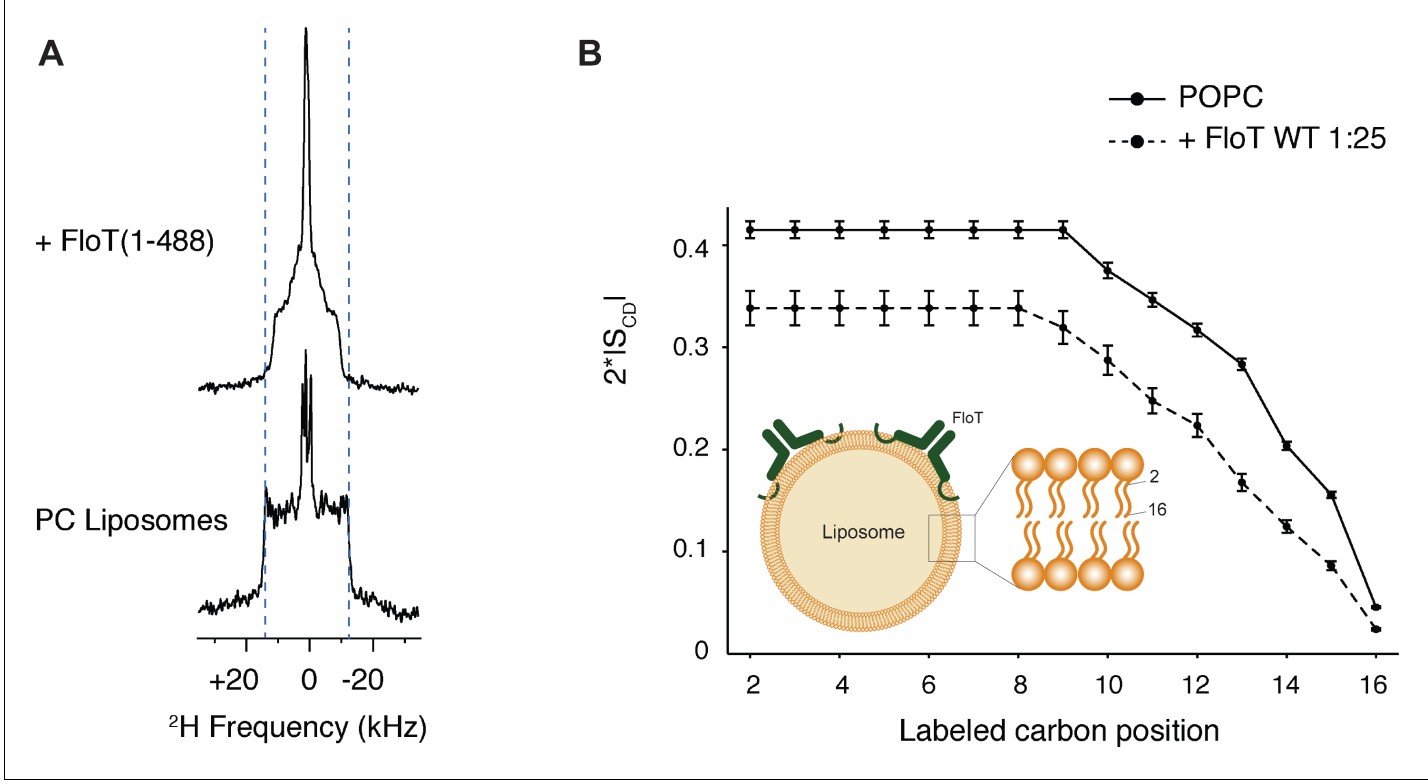

**Figure 5.** Lipid ordering of FloT probed by $^2$H solid-state NMR. (**A**) Wide-line $^2$H spectra of POPC-d31 liposomes with or without FloT at a lipid-to-protein molar ratio of 25:1 acquired at 298 K. (**B**) Effect of FloT on the C-$^2$H order parameters of the PC acyl chain. De-Pake-ing and simulations were applied on the $^2$H solid-state NMR spectra to determine accurately individual quadrupolar splittings. Order parameters of POPC-d31 acyl chain were derived from experimental quadrupolar splittings and plotted as a function of the labelled carbon position. *Insert:* schematic depiction of a liposome with added FloT which attaches to the membrane via a hairpin loop (**Bach and Bramkamp, 2015**).

The online version of this article includes the following figure supplement(s) for figure 5:

**Figure supplement 1.** $^{31}$P solid-state NMR experiments of POPC liposomes with or without FloT at a lipid-to-protein molar ratio of 25:1.

has an important impact on the order parameter along the entire acyl chain. It is remarkable that the protein significantly decreases the order parameter $S_{CD}$, reaching even the inner carbon atoms of the acyl chain, indicating a different packing behaviour and increased membrane fluidity upon interaction with FloT (*Figure 5B*). The strong fluidising effect described for FloT is notably different from the effects other proteins have when reconstituted into liposomes, such as plant remorins (*Legrand et al., 2019*) or the membrane binding peptide of the nonreceptor tyrosine kinase Src (*Scheidt and Huster, 2009*). The anisotropic lineshape of the $^{31}$P spectra indicates that the membrane is in the lamellar phase as expected for POPC at the chosen temperature (298K) (*Huster, 2014*). Upon interactions with FloT the lamellar phase remains intact with formation of a few smaller objects, indicating that the overall liposome structure is not affected and that its phase is maintained (*Figure 5—figure supplement 1*).

## Discussion

Our data provide evidence that flotillins play a direct role in controlling membrane fluidity and that membrane fluidity is critical for peptidoglycan synthesis at certain growth conditions. In vitro, flotillins enhance the fluidity of a model membrane, and in vivo, the membranes of fast growing flotillin-mutant cells are less fluid even though the fatty acid composition in these cells is identical. Therefore, we propose that the effect of flotillins on membrane fluidity is direct, through a change in the packing behaviour of the lipids resulting in an efficient separation of states of liquid ordered and disordered lipid domains in the membrane bilayer (*Bach and Bramkamp, 2013*). We found that

membrane fluidity is not solely a function of temperature, but also of growth conditions. In vivo, flotillins may also recruit specific, more rigid lipids, such as hopanoids and carotenoids (*Bramkamp and Lopez, 2015*; *García-Fernández et al., 2017*; *López and Kolter, 2010*) which have been found in association with FMMs, and whose synthesis could be growth condition dependent. The predominantly physical role in membrane organisation for flotillins fits with our observation that adding a chemical fluidiser is sufficient to restore MreB dynamics and cell shape to fast growing cells that lack flotillins. We propose that in fast-growing cells on rich medium, flotillin-mediated control of membrane fluidity is critical and sufficient to allow essential membrane bound processes, such as peptidoglycan synthesis, to proceed normally.

A sufficiently fluid membrane is necessary for the efficient recruitment and movement of MreB, and provides a more favourable environment for the peptidoglycan precursor LipidII (*Hussain et al., 2018*; *Ursell et al., 2014*; *Schirner et al., 2015*). It has recently been shown that modulation of either MreBCD or PBP1 levels is sufficient to alter the shape of *B. subtilis* cells (*Dion et al., 2019*), underscoring the importance of both systems. In the absence of flotillins, the activity of the MreBCD component is strongly reduced – as evidenced by the reduction of MreB speed – and the overall rigidity of the membrane is increased. This results in a less favourable environment for the peptidoglycan precursor LipidII, which prefers more liquid, disordered membrane phases (*Ganchev et al., 2006*; *Witzke et al., 2016*; *Calvez et al., 2019*). Our data indicate that the reduction in elongasome activity, which does not impact the growth rate itself, is compensated by increased peptidoglycan synthesis activity around division sites in flotillin mutants, which is sufficient to keep the overall cell shape intact, although cells are elongated. The accumulation of lipid dyes indicative of increased fluidity at division sites is in line with a recent study that showed phases of different fluidity in *Streptococcus pneumoniae* membranes, with more fluid membranes and LipidII localising at midcell (*Calvez et al., 2019*) where the membrane is most bent. Our findings are also in agreement with the recent observation that *B. subtilis* cells elongate and lose organisation of MreB when membrane fluidity is decreased by altering the membrane fatty acid composition (*Gohrbandt, 2019*) (H. Strahl, personal communication). It could very well be that the shift of fluidity towards the septum is only relative as the overall fluidity of the membrane is decreased in the absence of flotillins. This is yet to be determined, as the resolution of Laurdan imaging does not allow conclusive statements about local fluidity changes at the septum. The observation that reduced MreB mobility and therefore altered lateral cell wall synthesis lead to accumulation of Van-FL and HADA staining at the septum is not immediately conclusive. Since septal PG synthesis is MreB independent in *B. subtilis*, a direct effect of MreB seems unlikely. Rather, a reduction of overall membrane fluidity in a flotillin knock-out might impair LipidII dynamics within the membrane. MurG is the enzyme that catalyses the final step of LipidII synthesis. There are several reports that MurG localises to the septum in different organisms (*Aaron et al., 2007*; *Mohammadi et al., 2007*). Thus, it seems likely that the septum is a place of increased LipidII synthesis and a change in membrane fluidity would create problems for LipidII molecules to diffuse away from their insertion site, resulting in reduced lateral PG synthesis and MreB mobility. Alternatively, the reduced MreB mobility and reduced lateral PG synthesis lead to a reduced LipidII consumption at the lateral wall, and the excess LipidII is used by the septal PG synthesis machinery, thereby leading to an increase in midcell PG. It remains to be tested which of these possibilities is responsible for the observed phenotype. Nevertheless, in both cases the increase in cell wall staining at the septum would be indicative of a higher local synthesis activity.

It may seem paradoxical that cells elongate when elongasome activity is reduced, but one has to remember that a large amount of the peptidoglycan synthesis contributing to elongation of bacterial cells is actually taking place at midcell, before the ingrowth of the septum (*Aaron et al., 2007*; *Pazos et al., 2018*; *Varma and Young, 2009*). A relative increase in peptidoglycan synthesis at future division sites makes the activity of PBP1 critical and explains why its deletion has such a dramatic effect in cells lacking flotillins. Restoring fluidity using a chemical fluidiser allows the MreBCD component to again efficiently drive peptidoglycan synthesis during elongation, which is sufficient to suppress the flotillin mutant phenotype. The net effect of this is that cells lacking PBP1 and flotillins grown with benzyl alcohol behave as cells that only lack PBP1, which is quite similar to wild type. At low growth rates, there is no difference between wild type cells and cells lacking flotillins with respect to membrane fluidity, and the speed of MreB is similar between the two cell types. Thus the deletion of flotillins does not exacerbate the phenotype of cells lacking PBP1. The reason for the change in membrane fluidity between cells grown on rich or minimal medium is not yet clear – it

does not seem to be caused by a large shift in the fatty acid composition of the membranes. Various factors could play a role, such as the synthesis of specific lipids (hopanoids, isoprenoids) on either type of medium, but also protein crowding, which is higher in membranes of fast-growing cells than in slow-growing cells (*Szenk et al., 2017*). It will be an important future challenge to establish the cause for this difference. An overall rigidification of the membrane may also lead to retardation of processes which require membrane modifications such as division and sporulation, which is indeed observed in *B. subtilis* flotillin mutants (*Dempwolff et al., 2012*; *Donovan and Bramkamp, 2009*).

One of the proposed roles for flotillin proteins is that they form a 'platform' that transiently interacts with membrane proteins that need to oligomerise into functional complexes (*Lopez and Koch, 2017*). We tested this hypothesis for *B. subtilis* PBPs by comparing their localisation and oligomerisation in wild type and flotillin mutant strains. Although we were capable of detecting a high MW complex containing various PBPs (notably PBP1, 2a, 2b, 3 and 4), the complex was not dependent on the presence of flotillins. We also note that PBP1 was present in the complex, as well as present in a large smear in the first dimension native gel, which would explain why PBP1 was detected by mass spectrometry analysis of a native PAGE band containing FloA (*Schneider et al., 2015a*). PBP5, on the other hand, was not part of the high MW complex, which fits with its role in processing of the terminal D-Ala from stem-peptides that have not been cross-linked, which it exercises over the entire surface of the cell (*Kuru et al., 2012*). Although we cannot exclude that flotillins may affect PBPs that are not easily detected by Bocillin-FL, our results do not provide any evidence for a role for flotillins in the oligomerisation of PBPs in *B. subtilis*. This extends the finding of the Graumann lab that found either transient or no colocalisation between flotillins and other proteins present in DRM fractions (*Dempwolff et al., 2016*). Although it is obvious that peptidoglycan synthesis is altered in the absence of flotillins, our data strongly suggest that the basis for this alteration is in the physical organisation of the membrane rather than inefficient formation of divisome or elongasome complexes in the absence of flotillins, because flotillin mutants strains show no synthetic phenotype on minimal medium, and the defects on rich medium can be reverted by chemically fluidising the membrane.

In conclusion, our data provide a new model for flotillin function in the physical organisation of membranes during fast growth. The observation that flotillins differentially affect the membrane in different growth conditions also explains the diversity of phenotypes described for flotillin mutants in the literature.

## Materials and methods

**Key resources table**

| Reagent type (species) or resource | Designation | Source or reference | Identifiers | Additional information |
|---|---|---|---|---|
| Strain, strain background (*Escherichia coli*) | BL21(DE3) | Thermo Fisher Scientific | EC0114 | Chemically competent cells |
| Strain, strain background (*Bacillus subtilis*) | BB001 | *Bach and Bramkamp, 2013* | trpC2 yqfA::tet | |
| Strain, strain background (*Bacillus subtilis*) | BB003 | *Bach and Bramkamp, 2013* | trpC2 yuaG::pMUTIN4 yqfA::tet | |
| Strain, strain background (*Bacillus subtilis*) | DB003 | *Donovan and Bramkamp, 2009* | trpC2 yuaG::pMUTIN4 | |
| Strain, strain background (*Bacillus subtilis*) | RWBS5 | *Domínguez-Escobar et al., 2011* | trpC2 amyE::spc $P_{xyl}$-mrfpruby-mreB | |
| Strain, strain background (*Bacillus subtilis*) | PS832 | *Popham and Setlow, 1995* | Prototrophic revertant of 168 | |

*Continued on next page*

Continued

| Reagent type (species) or resource | Designation | Source or reference | Identifiers | Additional information |
|---|---|---|---|---|
| Strain, strain background (*Bacillus subtilis*) | 2082 | *Scheffers et al., 2004* | trpC2 pbpD::cat $P_{xyl}$–gfp–pbpD $^{1-510}$ | |
| Strain, strain background (*Bacillus subtilis*) | 2083 | *Scheffers et al., 2004* | trpC2 ponA::cat $P_{xyl}$–gfp–ponA $^{1-394}$ | |
| Strain, strain background (*Bacillus subtilis*) | 2085 | *Scheffers et al., 2004* | trpC2 dacA::cat Pxyl–gfp–dacA $^{1-423}$ | |
| Strain, strain background (*Bacillus subtilis*) | 3105 | *Scheffers et al., 2004* | trpC2 pbpC::cat Pxyl-gfp–pbpC $^{1-768}$ | |
| Strain, strain background (*Bacillus subtilis*) | 3122 | *Scheffers et al., 2004* | trpC2 pbpB::cat $P_{xyl}$-gfp-pbpB $^{1-825}$ | |
| Strain, strain background (*Bacillus subtilis*) | 3511 | *Scheffers and Errington, 2004* | trpC2 ponA::spc | |
| Strain, strain background (*Bacillus subtilis*) | 4042 | *Lages et al., 2013* | trpC2 pbpA::cat $P_{xyl}$-mkate2-pbpA $^{1-804}$ | |
| Strain, strain background (*Bacillus subtilis*) | 4056 | *Morales Angeles et al., 2017* | trpC2 dacA::kan | |
| Strain, strain background (*Bacillus subtilis*) | 4059 | This work | trpC2 dacA::cat $P_{xyl}$-gfp–dacA $^{1-423}$ yuaG::pMUTIN4 yqfA::tet | Scheffers lab |
| Strain, strain background (*Bacillus subtilis*) | 4064 | This work | trpC2 dacA::kan yuaG::pMUTIN4 yqfA::tet | Scheffers lab |
| Strain, strain background (*Bacillus subtilis*) | 4090 | This work | trpC2 ponA::spc yuaG::pMUTIN4 | Scheffers lab |
| Strain, strain background (*Bacillus subtilis*) | 4091 | This work | PS832 ponA::spc yqfA::tet | Scheffers lab |
| Strain, strain background (*Bacillus subtilis*) | 4092 | This work | trpC2 ponA::spc yuaG::pMUTIN4 yqfA::tet | Scheffers lab |
| Strain, strain background (*Bacillus subtilis*) | 4095 | This work | trpC2 ponA::cat $P_{xyl}$-gfp–ponA $^{1-394}$ yuaG::pMUTIN4 yqfA::tet | Scheffers lab |
| Strain, strain background (*Bacillus subtilis*) | 4099 | This work | trpC2 pbpB::cat Pxyl-gfp-pbpB $^{1-825}$ yuaG::pMUTIN4 yqfA::tet | Scheffers lab |
| Strain, strain background (*Bacillus subtilis*) | 4102 | This work | trpC2 pbpA::cat Pxyl-mkate2-pbpA $^{1-804}$ yuaG::pMUTIN4 yqfA::tet | Scheffers lab |
| Strain, strain background (*Bacillus subtilis*) | 4108 | This work | trpC2 pbpD::cat $P_{xyl}$-gfp–pbpD $^{1-510}$ yuaG::pMUTIN4 yqfA::tet | Scheffers lab |

*Continued*

| Reagent type (species) or resource | Designation | Source or reference | Identifiers | Additional information |
|---|---|---|---|---|
| Strain, strain background (*Bacillus subtilis*) | 4122 | This work | trpC2 pbpC::cat $P_{xyl}$-gfp–pbpC $^{1–768}$ yuaG::pMUTIN4 yqfA::tet | Scheffers lab |
| Strain, strain background (*Bacillus subtilis*) | 4128 | This work | trpC2 ponA::spc pbpD::cat $P_{xyl}$-gfp–pbpD $^{1–510}$ | Scheffers lab |
| Strain, strain background (*Bacillus subtilis*) | 4129 | This work | trpC2 ponA::spc yuaG::pMUTIN4 pbpD::cat $P_{xyl}$-gfp–pbpD $^{1–510}$ | Scheffers lab |
| Strain, strain background (*Bacillus subtilis*) | 4070 | This work | trpC2 mreB::kan amyE::spc $P_{xyl}$-mrfpruby-mreB | Scheffers lab |
| Strain, strain background (*Bacillus subtilis*) | 4076 | This work | trpC2 mreB::kan amyE::spc $P_{xyl}$-mrfpruby-mreB yuaG::pMUTIN4 yqfA::tet | Scheffers lab |
| Strain, strain background (*Bacillus subtilis*) | 4259 | This work; Veening et al., 2009 | trpC2 amyE::$P_{rrnB}$-gfp | Scheffers lab |
| Other | Bocillin | Thermo Fisher Scientific | BOCILLIN FL Penicillin, Sodium Salt | 5 µg/ml |
| Other | HADA | Synthesised as described (Morales Angeles et al., 2017) | 7-hydroxycoumarin 3-carboxylic acid-amino-D-alanine | 50 µM |
| Other | Vancomycin-FL | Sigma-Aldrich and Molecular Probes (Zhao et al., 2017) | Van-FL | 1:1 mixture of Vancomycin and BODIPYFL Vancomycin (Zhao et al., 2017), final concentration 1 µg/ml |
| Other | Nile Red | Thermo Fisher Scientific | 5H-Benzo[α]phenoxazin-5-one, 9-(diethylamino)-7385-67-3 | 0.5 µg/ml |
| Other | 16:0-d31-18:1 PC | Avanti | 860399 | Phospholipids |
| Other | Laurdan | Sigma-Aldrich | 6-Dodecanoyl-N,N-dimethyl-2-naphthylamine | - |
| Other | Benzyl alcohol | Sigma-Aldrich | Benzyl alcohol | - |
| Other | DiI-C12 | Thermo Fisher Scientific | 1,1'-Didodecyl-3,3,3',3'-Tetramethylindocarbocyanine Perchlorate | 2.5 µg/ml |
| Other | FM4-64 | Thermo Fischer Scientific | (N-(3-Triethylammoniumpropyl)—4-(6-(4-(Diethylamino) Phenyl) Hexatrienyl) Pyridinium Dibromide) | 0.5 µg/ml, Invitrogen FM 4–64 Dye |
| Software, algorithm | Prism 5 | 1992–2010 GraphPad Software | RRID:SCR_002798 | - |

*Continued on next page*

*Continued*

| Reagent type (species) or resource | Designation | Source or reference | Identifiers | Additional information |
|---|---|---|---|---|
| Software, algortihm | ImageJ 1.52p/FIJI | Wayne Rasband – National Institutes of Health, USA | RRID:SCR_002285 | Free software |
| Software, algorithm | SPSS | SPSS | RRID:SCR_002865 | software |
| Software, algorithm | NMR Depaker 1.0rc1 software | [Copyright (C) 2009 Sébastien Buchoux] | | software |
| Software, algorithm | Bruker Topspin 3.2 software | Bruker | RRID:SCR_014227 | software |

## *B. subtilis* strains and growth conditions

All *B. subtilis* strains used in this study are derived from strain 168 and are listed in the Key resources table. Construction of new strains was based on natural competence of *B. subtilis* (*Harwood and Cutting, 1990*). Gene integration or deletion was validated by colony PCR whereas the expression and localisation of the fluorescent fusions was additionally validated by microscopy. Cells were grown either in LB Lennox (5 g/L yeast extract; 5 g/L NaCl; 10 g/L tryptone) (*Lennox, 1955*) or Spizizen minimal medium (SMM) (*Anagnostopoulos and Spizizen, 1961*) supplemented with 1% glucose, at 37°C and 200 rpm, unless indicated otherwise. Induction of the $P_{xyl}$ promoter was triggered by addition of 0.2–0.5% xylose. Cell cultures were supplemented with spectinomycin (50 µg/ml), tetracycline (10 µg/ml), chloramphenicol (5 µg/ml), kanamycin (5 µg/ml), erythromycin (1 µg/ml), benzyl alcohol (BnOH, 0.1%) or magnesium sulphate ($MgSO_4$, 6–20 mM) when necessary.

## Growth curves

Growth experiments were performed either manually or automatically with a PowerWave 340 microplate reader (BioTek Instruments, U.S.A). Strains were pre-cultured overnight in 3 ml LB or SMM medium at 37°C with shaking at 200 rpm. Next, stationary or late-exponentially cells were diluted with fresh LB or SMM medium (supplemented when necessary), and cell densities ($OD_{600}$) were measured every 1 hr when monitored manually or every 10 min when monitored automatically, for a total time of 7–22 hr.

## Fluorescence microscopy

For standard fluorescence microscopy, exponentially growing cells were immobilised on microscope slides covered with a thin film of 1% agarose (w/v) in water or the appropriate medium. For TIRFM, agarose pads were mounted using Gene Frames (1.7 × 2.8 cm chamber, 0.25 mm thickness, 125 µL volume) from ThermoScientific. Standard fluorescence microscopy was carried out using an Axio Zeiss Imager M1 fluorescence microscope (EC Plan-Neofluar 100x/1.30 Oil Ph3 objective) equipped with an AxioCam HRm camera and an Nikon-Ti-E microscope (Nikon Instruments, Tokyo, Japan) equipped with Hamamatsu Orca Flash 4.0 camera.

For Laurdan and TIRFM experiments, a Delta Vision Elite microscope (Applied Precision, GE Healthcare) equipped with an Insight SSI Illumination, an X4 Laser module, a CoolSnap HQ (*Zhao et al., 2017*) CCD camera and a temperature-controlled chamber set up at 37 $^0$C was used. Laurdan images were taken with an Olympus UplanSApo 100x/1.4 oil objective. TIRFM image series were taken using an Olympus UAPO N 100X/1.49 TIRF objective and a 561 nm laser (50 mW, 100% power). Data processing was performed with softWoRx Suite 2.0 Software.

## Visualisation of cell wall synthesis

Peptidoglycan (PG) synthesis was assessed by labelling the cells with HADA (7-hydroxycoumarin 3-carboxylic acid-amino-D-alanine) (*Kuru et al., 2015*), Van-FL (*Daniel and Errington, 2003*) or D-Ala-D-Pra (*Sarkar et al., 2016*).

HADA: synthesised as described (*Morales Angeles et al., 2017*). Overnight cultures of *B. subtilis* strains were diluted 1:100 into fresh LB medium or LB medium supplemented with 0.1% (w/v) of

benzyl alcohol (BnOH), a membrane fluidiser. Cells were grown until exponential phase, a sample of 1 ml of culture was spun down for 30 s, 5000 × g and the cell pellet was resuspended in 25 µl of fresh pre-warmed LB or LB containing 0.1% (w/v) BnOH. HADA was added to a 50 µM final concentration. Cells were incubated for 10 min in the dark (37 $^0$C, 200 rpm) and then washed twice in 1 ml PBS buffer (58 mM Na$_2$HPO$_4$, 17 mM NaH$_2$PO$_4$, 68 mM NaCl, pH 7.3) to remove the excess of unbounded HADA. Cells were spun down again and resuspended in 25 µl of the appropriate medium and 2 µl of cells were mounted on 1% agarose slides before visualisation. Visualisation of HADA patterns (excitation: 358 nm/emission: 461 nm) under fluorescence microscopy from two biological replicates and cell length measurements were taken from at least 100 cells each strain/treatment.

Van-FL: a 1:1 mixture of vancomycin (Sigma Aldrich) and BODIPYFL Vancomycin (Molecular Probes) at a final concentration 1 µg/ml was used to label cells for 5–10 min at room temperature.

D-Ala-D-Pra: synthesised as described (*Sarkar et al., 2016*). 1 ml of cell cultures was pelleted and resuspended in 50 µl of PBS buffer. Dipeptide was added to a final concentration of 0.5 mM following with 5 min incubation at room temperature. Cells were fixed by adding 70% ethanol and incubated for minimum 2 hr in −20˚C. Next, cells were washed twice with PBS in order to remove unattached peptides, and resuspended in 50 µl of PBS. The D-Ala-D-Pra was subsequently labelled via a click reaction with fluorescent azide (20 µM) that was incubated for 15 min at room temperature with addition of copper sulphate (CuSO4, 1 mM), tris-hydroxypropyltriazolylmethylamine (THPTA, 125 µM) and ascorbic acid (1.2 mM). The sample was washed twice with PBS and resuspended in 50 µl of the same buffer.

## Laurdan staining and GP calculations

Laurdan (6-Dodecanoyl-N, N-dymethyl2-naphthylamine, Sigma-Aldrich) was used to detect the liquid ordering in the membrane, as decribed (*Bach and Bramkamp, 2013*), with modifications. Cells were grown in LB or SMM medium until late exponential phase. Laurdan, dissolved in dimethyformamide (DMF), was added at 10 µM final concentration and cells were incubated for 10 min in the dark at 37 ˚C, 200 rpm. Cells were then washed twice in PBS buffer supplemented with 0.2% (w/v) glucose and 1% (w/v) DMF, and resuspended in fresh prewarmed appropriate medium. Laurdan was excited at 360 ± 20 nm, and fluorescence emission was captured at 460 ± 25 nm (exposure time: 500 ms) and at 535 ± 25 nm (exposure time: 1 s) (*Strahl et al., 2014*). The image analysis including the generation of GP maps was carried out using Fiji Software (*Schindelin et al., 2012*) in combination with the macro tool CalculateGP designed by Norbert Vischer (http://sils.fnwi.uva.nl/bcb/objectj/examples/CalculateGP/MD/gp.html). The GP values were measured for at least 100 individual cells after background subtraction, from two biological replicates.

## Other fluorescent membrane probes

*B. subtilis* cell membranes were probed with Nile Red (0.5 µg/ml), FM4-64 (0.5 µg/ml) or DiI-C12 (2.5 µg/ml). To this end, an overnight culture was grown in appropriate antibiotic, diluted 1:100 in LB medium supplemented with DiI-C12 followed by growth until exponential phase. Membranes were probed with Nile Red or FM4-64 for 5 min at room temperature after reaching exponential phase. The stained cells were washed three times in prewarmed LB medium supplemented with 1% DMSO before visualisation under fluorescence microscopy.

## TIRF time lapse microscopy

Time-lapse TIRFM movies were taken in two independent experiments for each strain and condition. To this end, overnight cultures of strains grown in LB medium supplemented with the appropriate antibiotic were diluted 1:100 in medium containing 0.5% (w/v) xylose and grown until exponential phase. All experiments were performed inside the incubation chamber set to 37 ˚C, no longer than 10 min after taking the sample. The cells were imaged over 30 s with 1 s inter-frame intervals in a continuous illumination and ultimate focus correction mode. The single particle tracking analyses and kymographs were done using Fiji Software (*Schindelin et al., 2012*) in combination with the MTrackJ (*Meijering et al., 2012*) and MicrobeJ plugins (*Ducret et al., 2016*).

## Bocillin labelling

Cells were grown until an OD600 of 0.4–0.5, and washed twice with PBS. Next, samples were resuspended in 50 μl PBS containing Bocillin-FL (5 μg/ml) and incubated at room temperature for 10 min. Subsequently cells were harvested, lysed by sonication and cell-free extracts were prepared. Samples, equalised for culture OD, were prepared with SDS-PAGE sample buffer and run on a 12% SDS-PAGE gel. Fluorescent bands were visualised using a Typhoon Trio (GE Healthcare) scanner.

## Isolation of membranes

Membrane isolation was adapted from *Schneider et al., 2015b*. Briefly, cells were grown until an OD600 of 0.4–0.5, cell fractions were collected and resuspended in PBS with Lysozyme (1 μg/ml), EDTA (5 mM), 1/10 tablet cOmplete protease inhibitor (Roche), and DNAse (5 μg/ml) and incubated for 30 min on ice. Samples were sonicated, cell which did not lyse were spun down (8000 rpm, 2 min, 4°C), and the supernatant fraction was centrifuged at 4°C and 40000 rpm for 1 hr. The membrane pellet was dissolved in ACA750 buffer (750 mM aminocaproic acid, 50 mM Bis-Tris, pH 7.0) to a final protein concentration of 1 μg/μl. Membranes were solubilised overnight at 4°C in 1% (w/v) dodecylmaltoside (DDM) and either used directly or stored at −20°C.

## Blue native PAGE (BN-PAGE)

The experiment was performed as described (*Trip and Scheffers, 2016*). Samples were prepared by mixing sample buffer (0.1% Ponceau S, 42.5% Glycerol) with solubilised membranes in a 1:3 ratio. Samples were resolved on a mini-PROTEAN TGX Stain-Free gradient gel (4–15%, BioRad) using cathode (50 mM Tricine and 15 mM BisTris), and anode (50 mM BisTris pH 7.0) buffers. The Novex NativeMark Unstained Protein Standard marker was used as a Mw marker.

## Second dimension SDS PAGE (2D SDS-PAGE)

A lane of interest was excised from the Native-PAGE gel and immobilised horizontally on top of a SDS-PAGE gel (5% stacking, 12% resolving). The excised fragment was flanked with a piece of Whatman paper soaked with PageRuler Prestained Protein Ladder. The gel fragment to be resolved in the second dimension was topped with a mix of 1% (w/v) LowTemperature agarose, 0.5% (w/v) SDS and bromophenol blue. After the agarose had solidified, standard SDS-PAGE electrophoresis was performed.

## TEM

Cultures were harvested by centrifugation and a small amount of pellet was placed on a copper dish. A 400 copper mesh grid and a 75 μm aperture grid was placed on top of the cells to create a thin layer. The sandwiched cells were plunged rapidly into liquid propane. Sandwiches were then disassembled and placed on frozen freeze-substitution medium containing 1% osmium tetroxide, 0.5% uranyl acetate and 5% water in acetone. Cells were dehydrated and fixed using the rapid freeze substitution method (*McDonald, 2014*). Samples were embedded in epon and ultrathin sections were collected on formvar coated and carbon evaporated copper grids and inspected using a CM12 (Philips) transmission electron microscope. For each strain 70 random septa were imaged with pixel resolution of 1.2 nm. Using ImageJ the cell wall thickness for each septum was measured at 4 places from which the average was taken.

## Statistical analysis

Each set of micrographs to be analysed was imaged with the same exposure time. For the septum intensity analysis of HADA, FM4-64 and Nile Red, the wildtype strain (expressing GFP) and the Δ*floAT strains* were mixed, labelled and imaged on the same agarose pad. Intensity of the fluorescently labelled septa was measured using the ObjectJ macro tool PeakFinder (https://sils.fnwi.uva.nl/bcb/objectj/examples/PeakFinder/peakfinder.html) (*Vischer et al., 2015*). A perpendicular line was drawn across the septal plane, the background intensity was removed resulting in a maximum peak intensity. The number of septa compared was indicated for every individual experiment. Populations were compared using the non-parametric Mann-Whitney test. The null hypothesis was tested with the $p$ value of 0.05. The statistical analyses and their graphical representation (box plots) were generated with GraphPad Prism 8.1 (San Diego, California, USA). Box plots show the median and

the interquartile range (box), the 5th and 95th percentile (whiskers). Laurdan fluorescence generalised polarisation, cell length and MreB speed statistical analyses were performed using Kruskal-Wallis with Dunn's multiple comparison *post-hoc* test.

## Fatty acid composition analysis

The fatty acid composition of *B. subtilis* wild-type cells and the flotillin/PBP mutants was analysed with gas chromatography as fatty acid methyl esters. Cells for the analyses were grown at 37°C in LB or SMM until mid-exponential ($OD_{600}$ ~0.5), harvested (6000 rpm, 10 min, 4°C) and washed with 100 mM NaCl. Next, the cells were freeze dried at −50°C, 0.012 mbar for a minimum of 18 hr. All analyses were carried out on biological duplicates by the Identification Service of the DSMZ, Braunschweig, Germany.

## Sample preparation for solid-state NMR

FloT was essentialy purified as described (*Bach and Bramkamp, 2013*), in solubilised form, and stored in buffer A (50 mM Tris HCl pH 7.5, 150 mM NaCl, 5 mM $MgCl_2$) supplemented with 0.05% Triton X-100.

Liposomes containing POPC-d31 were prepared by mixing appropriate lipid powders in organic solvents (chloroform/methanol, 2:1 ratio). Solvents were evaporated under a flow of $N_2$ to obtain a thin lipid film. Lipids were rehydrated with ultrapure water before lyophilisation over night. The lipid powder was hydrated with an appropriate amount of buffer A with 10% glycerol and homogenised by three cycles of vortexing, freezing (liquid nitrogen, −196°C, 1 min) and thawing (40°C in a water bath, 10 min). This protocol generated a milky suspension of micrometer-sized multilamellar vesicles. FloT was solubilised in Buffer A supplemented with 0.05% Triton X-100 and added to preformed liposomes and incubated for 1 hr at room temperature. A dialysis step was then performed against Buffer A at 4°C under agitation to remove the detergent. Samples were centrifuged at 100,000 g at 4°C for 1 hr to pellet the proteoliposomes. $^2$H solid-state NMR spectra were recorded of liposomes in the presence or absence of FloT at a lipid/protein ratio of 25:1 at 298 K.

## Solid-state NMR

$^2$H NMR spectroscopy experiments were performed using a Bruker Avance III 500 MHz WB (11.75 T) spectrometer. They were recorded on $^2$H-labelled POPC at 76.77 MHz with a phase-cycled quadrupolar echo pulse sequence (90°x-t-90°y-t-acq). Acquisition parameters were as follows: spectral window of 500 kHz for 2H NMR spectroscopy, p/2 pulse width of 3.90 ms for $^2$H, interpulse delays (t) were of 40 μs, recycled delays of 1.3 s for 2H; 3000 and 8000 scans were used for $^2$H NMR spectroscopy on liposomes and liposomes with FloT, respectively. Spectra were processed using a Lorentzian line broadening of 300 Hz for $^2$H NMR spectra before Fourier transformation from the top of the echo. Samples were equilibrated for 30 min at a given temperature before data acquisition. All spectra were processed and analysed using Bruker Topspin 3.2 software. Spectral moments were calculated for each temperature using the NMR Depaker 1.0rc1 software [Copyright (C) 2009 Sébastien Buchoux]. Orientational order parameters (SCD) were calculated from experimental quadrupolar splittings (DnQ) as described in *Huster, 2014*. For $^{31}$P ssNMR, we applied a static Hahn spin echo sequence at the $^{31}$P frequency of 162 MHz on a 400 MHz (9.4T) Bruker Avance III HD spectrometer, with a 90∘ pulse of 8 μs, a delay of 40 μs, a recycle delay of 5 s, a spectral window of 400 ppm and a number of scans of 4000 and 3400 was used on liposomes and liposomes with FloT, respectively. Spectra were processed using a Lorentzian line broadening of 100 Hz.

## Acknowledgements

We thank Henrik Strahl for discussions and sharing of unpublished data, Rut Carballido-Lopez for strain RWBS5 and Luiza Morawska and Oscar Kuipers for the P*rrnB-gfp* plasmid.

This work was funded by NWO grant 864.09.010 (DJS), DFG grants BR 2915/4–1; INST 86/1452–1 (MB), ERC starting grant 757913; NWO grant 721.014.008 (AKHH); PhD fellowships DAAD-GSSP to AS; and SFRH/BD/78061/2011- POPH/FSE/FCT to ASB.

## Additional information

### Funding

| Funder | Grant reference number | Author |
|---|---|---|
| Nederlandse Organisatie voor Wetenschappelijk Onderzoek | Vidi 864.09.010 | Dirk-Jan Scheffers |
| Deutsche Forschungsgemeinschaft | BR 2915/7-1 | Marc Bramkamp |
| Deutsche Forschungsgemeinschaft | INST 86/1452-1 | Marc Bramkamp |
| Nederlandse Organisatie voor Wetenschappelijk Onderzoek | 721.014.008 | Anna KH Hirsch |
| European Research Council | starting grant 757913 | Anna KH Hirsch |
| Deutscher Akademischer Austauschdienst | PhD fellowship DAAD-GSSP | Abigail Savietto |
| Fundação para a Ciência e a Tecnologia | PhD fellowship SFRH/BD/78061/2011- POPH/FSE/FCT | Anabela de Sousa Borges |

The funders had no role in study design, data collection and interpretation, or the decision to submit the work for publication.

### Author contributions

Aleksandra Zielińska, Conceptualization, Data curation, Formal analysis, Supervision, Investigation, Visualization, Methodology, Writing - original draft, Writing - review and editing; Abigail Savietto, Conceptualization, Data curation, Formal analysis, Investigation, Visualization, Methodology, Writing - original draft, Writing - review and editing; Anabela de Sousa Borges, Conceptualization, Funding acquisition, Investigation, Visualization, Methodology; Denis Martinez, Melanie Berbon, Formal analysis, Investigation, Methodology; Joël R Roelofsen, Investigation; Alwin M Hartman, Anna KH Hirsch, Resources; Rinse de Boer, Formal analysis, Investigation; Ida J Van der Klei, Supervision, Investigation; Birgit Habenstein, Conceptualization, Supervision, Investigation, Methodology, Writing - original draft; Marc Bramkamp, Dirk-Jan Scheffers, Conceptualization, Formal analysis, Supervision, Funding acquisition, Writing - original draft, Project administration, Writing - review and editing

### Author ORCIDs

Ida J Van der Klei ![iD] https://orcid.org/0000-0001-7165-9679
Marc Bramkamp ![iD] http://orcid.org/0000-0002-7704-3266
Dirk-Jan Scheffers ![iD] https://orcid.org/0000-0002-9439-9168

### Decision letter and Author response

Decision letter https://doi.org/10.7554/eLife.57179.sa1
Author response https://doi.org/10.7554/eLife.57179.sa2

## Additional files

### Supplementary files

• Transparent reporting form

### Data availability

All data generated or analysed during this study are included in the manuscript and supporting files.

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
