## [Decision Letter]

**Acceptance summary:**

In bacteria, Flotillins organize so-called micro-domains in the bacterial membrane that have been proposed to act as protein scaffolds for the local recruitment of protein complexes, including cell wall synthetic proteins. In this work, Zielinska et al. provide evidence for an alternative mechanism whereby Flotillins instead function to regulate fluidity of the membrane, indirectly affecting cytoskeletal dynamics and thus cell wall synthesis. In the future, it will be important to determine whether the proposed mechanisms are mutually exclusive or whether Flotillins indeed perform both functions.

**Decision letter after peer review:**

[Editors’ note: the authors submitted for reconsideration following the decision after peer review. What follows is the decision letter after the first round of review.]

Thank you for submitting your work entitled "Membrane fluidity controls peptidoglycan synthesis and MreB movement" for consideration by *eLife*. Your article has been reviewed by two peer reviewers, and the evaluation has been overseen by a Reviewing Editor and a Senior Editor. The reviewers have opted to remain anonymous.

Our decision has been reached after consultation between the reviewers. Based on these discussions and the individual reviews below, we regret to inform you that your work will not be considered for publication in *eLife* at this stage.

While the reviewers agree that the work conveys a potentially interesting message with regards to a new model of Flotillin function, they both indicate that the results are yet too preliminary to fully disprove the existing hypothesis that Flotillins form a physical platform for the recruitment of cell wall synthetic complexes. Specifically, the reviewers insist on two key points that need further investigations to strengthen the conclusions:

i) The quantification of Peptidoglycan and/or precursor accumulation at the septa is not supported by the EM images and Laurdan staining experiments. Comparison of fluorescence intensities between conditions is particularly delicate, requiring very precise microscope calibrations and possibly imaging of distinct cell genotypes in the same field of view. The conclusions that PG accumulation and membrane fluidity increase at septa in the floAt mutant is central to the authors' hypothesis and should be verified by a complementary approach.

ii) The benzyl alcohol rescue experiment is currently the main piece of evidence that challenges the previous model, but is it sufficient? Effects other than restoring membrane fluidity alone could explain growth restoration.

The policy at *eLife* is to reject manuscripts that cannot be revised within a three-months’ time period. If you think that you can address the comments in full and given the level of interest, we would be inclined to reconsider a profoundly revised manuscript, note however that it would be processed as a new submission and would have to go through another round of review.

Reviewer #1:

Zielinska and co-workers show here that flotillins affect overall membrane fluidity in *Bacillus subtilis* and that this explains their effect on the activity of MreB and peptidoglycan synthesis. They discovered this phenomenon by observing differences in cell wall synthesis at division sites and related differences in lipid staining in a flotillin deletion mutant. An important discovery was that they could exacerbate the phenotype by combining it with a deletion in the non-essential peptidoglycan synthase PBP1. What completed their experiments was the fact that they could restore the phenotype by fluidizing the cell membrane with an alcohol.

This is an interesting paper that highlights the importance of membrane fluidity, the role of bacterial flotillins in this, and the fact that this physiological characteristic changes under different growth conditions and can be adjusted by specific membrane proteins.

I agree with their overall conclusion but some conclusions seem to me incorrect (outlined below). The paper is clearly written, and experiments generally well performed. However, the microscopic figures are far too small. Even when displayed on a large screen many cellular details are not visible, e.g. Laurdan staining that does not show any clear septa. Firstly, a paper should be readable when printed, secondly, microscopic images should be clear so that phenotypic differences are clearly observable. This is currently not the case and should be improved.

I do not agree with the conclusion in the Discussion (second paragraph) that membrane fluidity and peptidoglycan synthesis is higher at septa in order to compensate the absence of flotillins. This reasoning does not make sense because what should be compensated for? The overall fluidity still changes. It is also a bit difficult how this can be compensated, in other words, how can the fluidity be increased at septa? Moreover, if there would be more peptidoglycan synthesis, then cells would divide faster, thus be shorter, since the authors show that septa do not become wider (Supplementary Figure 1). But a floAT deletion strain is not shorter (Figure 2). This discrepancy might be related to the possibility that the authors have not measured the septal intensities in Figure 1 correctly. That is, they should have compared the intensities of septa with that of the lateral wall, because it might well be that a floAT deletion strain has an overall more permeable cell wall that facilitate fluorophore diffusion. In fact the laurdan images in Figure 3 even suggest this. Maybe the authors measured the relative increase in intensity at septa, but I was unable to determine this from their experimental description. Moreover, and importantly, their laurdan images show no evidence for an increase of fluidity in septa. This seems to contradict their own argumentation.

Reviewer #2:

The experiments are well done and for the most part convincing. The problem is that their conclusions are overstated. The claim that all effects are explained by a direct action of flotillins on membrane fluidity rather than primarily affecting the assembly or activity of a PBP is not supported by the data. In fact most of the experiments (except potentially the benzyl alcohol addition, see below) can be explained by the floAT double mutant affecting primary septal PG synthesis or turnover. This does not mean that the manuscript is not suited for *eLife*, the data should just be interpreted differently.

1) One could state that all of the observed effects are due to floAT control of a septal PBP or septal PG remodeling. First, the claim that there is more lipidII/PG precursors in Figure 1 in the float mutants is based on a small difference and is weak. The signal is barely above background and HADA does not mark sites of PG synthesis, rather it marks sites of PG remodeling/turnover, as it depends on the exchange of the labeled D-ala in lipid II on the extracytoplasmic side. Moreover, the fact that floAT cells depend strongly on PBP1 for cell division, provides genetic evidence that the double mutant is under septal PG stress, which would probably be observable by applying PG stress in the cytoplasm (with Fosfomycin for example) and extracellularly with β-lactams or glycopeptides or bacitracin or A22.

If PBP biosynthetic activity is reduced (or PG degradation increased) in the floAT mutants, would this not lead to a (slight) accumulation of lipid II at the septum as well, and this could explain the observation in Figure 1? Also, this effect on PG stress could easily explain the reduction in MreB movement. Finally, my understanding was that an increase in lipidII would INCREASE membrane fluidity, but they show in Figure 3 that fluidity is reduced in the floAT mutant.

2) Growth rate effect: I disagree that the difference between LB and SMM can be pinned on growth rate. The cellular metabolism is completely different in minimal vs. rich medium so how can the authors rule out that a change in biosynthetic pathways does not affect lipid composition, PG (bactoprenol or lipidII) synthesis, isoprenoids or simply certain membrane proteins, all of which could alter membrane fluidity. The growth rate argument does not hold here, and it could only be proven if the authors stick to LB and lower growth rate there, ideally by reducing temp, but there could be complications on fluidity arising from a temperature shift. Inducing ppGpp would also be possibility, but this could also reprogram cell metabolism and make the interpretation ambiguous. Perhaps use diluted LB?

3) The benzyl alcohol addition experiment is the best (and in my opinion only) argument for their claim on membrane fluidity, but how do we know that this compound directly affects the membrane, rather than acting indirectly, like SMM, changing metabolism? How quick are the changes observed after addition of BA? Are the changes in fluidity and MreB localization still observed when cells are pre-treated with protein synthesis inhibitors and /or cerulenin (FA inhibitor)? By the way, are FRAP measurements of membrane-tethered GFP consistent with a change in fluidity?

The bottom line is: I'm not a fan of the previous flotillin model either, but I don't think the authors provide compelling evidence in favor of their model. In fact, their experimental data does not completely exclude the previous model about flotillins facilitating PG assembly/turnover machines either.

[Editors’ note: further revisions were suggested prior to acceptance, as described below.]

Thank you for submitting your article "Flotillin mediated membrane fluidity controls peptidoglycan synthesis and MreB movement" for consideration by *eLife*. Your article has been reviewed by two peer reviewers, and the evaluation has been overseen by a Reviewing Editor and Gisela Storz as the Senior Editor. The following individual involved in review of your submission has agreed to reveal their identity: Leendert Hamoen (Reviewer #2).

The reviewers have discussed the reviews with one another and the Reviewing Editor has drafted this decision to help you prepare a revised submission.

Overall the study is solid, convincing, well presented and certainly suitable for publication in *eLife*. Nevertheless, some clarifications should be provided before the manuscript is accepted for publication.

1) It is not clear why a reduction in MreB mobility in the membrane of floAT cells grown in LB would lead to increased vancomycin-FL and HADA staining at the septum. This relationship is poorly accounted for in the manuscript. MreB is not at the septum, so why would lower mobility of MreB cause an increase in septal lipid II? This observation is thus circumstantial. MreB and septal lipid II could be both affected because of reduced fluidity, but those events do not have to be linked:

– Perhaps membranes are less fluid and if lipid II is inserted at the septum, it is less likely to diffuse away in floAT cells because of the reduced fluidity?

– Or there is just a preference for lipid II to accumulate there because the membrane is curved and fluidity at the curvature is affected in the absence of FloAT? If that's the case, what does the additional deletion of pbp1 really tell us (although it is interesting)?

2) To clarify these issues, these questions should be discussed and it could be easier for the reader if the order of the results were turned around and tell the story more linearly. For example, rather than going from PG studies to membrane fluidity studies, the manuscript could present the established part first (NMR, fluidity, MreB) and then argue that reduced MreB mobility prompted imaging with HADA, Vanco-FL and finally check the pbp1floAT triple mutant. The current read takes a circular path (starting with PG and ending with MreB) and again there is no compelling causality between MreB movement and septal PG presented here.

---

## [Author Response]

[Editors’ note: the authors resubmitted a revised version of the paper for consideration. What follows is the authors’ response to the first round of review.]

While the reviewers agree that the work conveys a potentially interesting message with regards to a new model of Flotillin function, they both indicate that the results are yet too preliminary to fully disprove the existing hypothesis that Flotillins form a physical platform for the recruitment of cell wall synthetic complexes. Specifically, the reviewers insist on two key points that need further investigations to strengthen the conclusions:i) The quantification of Peptidoglycan and/or precursor accumulation at the septa is not supported by the EM images and Laurdan staining experiments. Comparison of fluorescence intensities between conditions is particularly delicate, requiring very precise microscope calibrations and possibly imaging of distinct cell genotypes in the same field of view. The conclusions that PG accumulation and membrane fluidity increase at septa in the floAt mutant is central to the authors' hypothesis and should be verified by a complementary approach.

We have carried out an additional series of microscopy experiments in which the distinct genotypes were imaged in the same field of view using two lipid dyes and one peptidoglycan synthesis dye. These results (more detail below) are completely in line with our initial findings and have now been included in the manuscript.

We would also like to note that the EM imaging and Laurdan staining are not suitable for the detection of precursor accumulation (or absence thereof) – and we make no statements that would suggest this. EM is used to measure septum thickness as a thicker septum might result in accumulation, Laurdan is used to determine overall membrane fluidity in cells and does not have the resolving power in our setup to say anything about differences at the septum.

We would like to emphasize that the main conclusion of this paper is the function of flotillins in fluidizing the membrane and the resulting effect on membrane coupled processes of which we use cell wall synthesis as one example.

ii) The benzyl alcohol rescue experiment is currently the main piece of evidence that challenges the previous model, but is it sufficient? Effects other than restoring membrane fluidity alone could explain growth restoration.

We fully agree that additional proof was necessary to prove the exact effects of flotillins on membranes. We have now included a biophysical experiment using state of the art solid state NMR that shows that the flotillin FloT alone is sufficient to fluidize membranes in model liposomes. This completely different approach provides an additional line of evidence that flotillins on their own are required and sufficient to fluidize the membrane.

Reviewer #1:Zielinska and co-workers show here that flotillins affect overall membrane fluidity in *Bacillus subtilis* and that this explains their effect on the activity of MreB and peptidoglycan synthesis. They discovered this phenomenon by observing differences in cell wall synthesis at division sites and related differences in lipid staining in a flotillin deletion mutant. An important discovery was that they could exacerbate the phenotype by combining it with a deletion in the non-essential peptidoglycan synthase PBP1. What completed their experiments was the fact that they could restore the phenotype by fluidizing the cell membrane with an alcohol.This is an interesting paper that highlights the importance of membrane fluidity, the role of bacterial flotillins in this, and the fact that this physiological characteristic changes under different growth conditions and can be adjusted by specific membrane proteins.I agree with their overall conclusion but some conclusions seem to me incorrect (outlined below). The paper is clearly written, and experiments generally well performed. However, the microscopic figures are far too small. Even when displayed on a large screen many cellular details are not visible, e.g. Laurdan staining that does not show any clear septa. Firstly, a paper should be readable when printed, secondly, microscopic images should be clear so that phenotypic differences are clearly observable. This is currently not the case and should be improved.

We thank the reviewer for the overall very positive comments. We have made adjustments to the figures to make them clearer. For easy reading we have inserted the figures in the text, but high resolution files are now also available that would be used for the published version.

We also want to point out that microscopic images showing laurdan measurement do not aim to highlight individual cells or even subcellular region (e.g.) septa. The resolution is not good enough to make strong claims about membrane fluidity measured by laurdan GP in distinct membrane regions. We provide an overview in a heat map coloration to support the calculated values shown in individual graphs. The micrographs allow the reader to quickly judge about the effect of flotillins deletions or benzyl alcohol addition on the overall membrane fluidity. We feel the provided images serve this idea perfectly.

I do not agree with the conclusion in the Discussion (second paragraph) that membrane fluidity and peptidoglycan synthesis is higher at septa in order to compensate the absence of flotillins. This reasoning does not make sense because what should be compensated for? The overall fluidity still changes. It is also a bit difficult how this can be compensated, in other words, how can the fluidity be increased at septa?

We agree that we cannot draw a definitive conclusion on the increase of fluidity at the septum in a flotillin deficient strain. Therefore, we have rephrased the statement in the Discussion to clearly indicate that it is speculative. We do note, however, that it is in line with other recent findings from the Zapun lab on increased fluidity at the division site.

The statement now reads:

“Our data indicate that the reduction in elongasome activity, which does not impact the growth rate itself, is compensated by a shift to increased peptidoglycan synthesis activity around division sites in flotillin mutants, which is sufficient to keep the overall cell shape intact, although cells are elongated. […] This is yet to be determined, as the resolution of Laurdan imaging does not allow conclusive statements about local fluidity changes at the septum.”

Moreover, if there would be more peptidoglycan synthesis, then cells would divide faster, thus be shorter, since the authors show that septa do not become wider (Supplementary Figure 1). But a floAT deletion strain is not shorter (Figure 2).

We respectfully disagree – increased peptidoglycan synthesis around the septum does not necessarily mean that all the peptidoglycan is contributing to the septum and allows faster division. In fact a recent preprint from the Deckers-Hebestreit and Strahl labs in which membrane fluidity is lowered by changing the fatty acid composition also shows a disruption of MreB activity (and thus lateral wall synthesis) while cells are slightly elongated (personal communication H. Strahl, length measurements were not provided in the preprint but do show elongation. Preprint: Biorxiv 10.1101/852160). It may seem paradoxical that cells elongate by synthesis around the division site but this is not uncommon, as is mentioned in the Discussion and backed up with various references.

This discrepancy might be related to the possibility that the authors have not measured the septal intensities in Figure 1 correctly. That is, they should have compared the intensities of septa with that of the lateral wall, because it might well be that a floAT deletion strain has an overall more permeable cell wall that facilitate fluorophore diffusion.

We have tackled this issue in two ways – first, we repeated a series of experiments with two membrane dyes and one PG dye using a wild type strain expressing GFP, which allowed mixing of the two strains and direct comparison of septal signal intensities to exclude the possibility that differences in microscopy setup cause the measured intensity difference (new Figure 1—figure supplement 1A, B). Second, we have excluded the possibility that diffusion is different between the strains. This cannot be done with a peptidoglycan synthesis dye as these could also be differently distributed because of differences in peptidoglycan. Thus, we compared intensities at the lateral wall for the membrane dye Nile Red, in the new experiment with the mixed strains. No difference in lateral intensity was observed indicating that there is no difference in fluorophore diffusion (new Figure 1—figure supplement 1D).

In fact the laurdan images in Figure 3 even suggest this. Maybe the authors measured the relative increase in intensity at septa, but I was unable to determine this from their experimental description. Moreover, and importantly, their laurdan images show no evidence for an increase of fluidity in septa. This seems to contradict their own argumentation.

Laurdan is used to determine overall membrane fluidity in cells and does not have the resolving power in our setup to say anything about differences at the septum. We have added a statement to clarify this to the Discussion (second paragraph).

Reviewer #2:The experiments are well done and for the most part convincing. The problem is that their conclusions are overstated. The claim that all effects are explained by a direct action of flotillins on membrane fluidity rather than primarily affecting the assembly or activity of a PBP is not supported by the data. In fact most of the experiments (except potentially the benzyl alcohol addition, see below) can be explained by the floAT double mutant affecting primary septal PG synthesis or turnover. This does not mean that the manuscript is not suited for eLife, the data should just be interpreted differently.1) One could state that all of the observed effects are due to floAT control of a septal PBP or septal PG remodeling. First, the claim that there is more lipidII/PG precursors in Figure 1 in the float mutants is based on a small difference and is weak. The signal is barely above background and HADA does not mark sites of PG synthesis, rather it marks sites of PG remodeling/turnover, as it depends on the exchange of the labeled D-ala in lipid II on the extracytoplasmic side. Moreover, the fact that floAT cells depend strongly on PBP1 for cell division, provides genetic evidence that the double mutant is under septal PG stress, which would probably be observable by applying PG stress in the cytoplasm (with Fosfomycin for example) and extracellularly with β-lactams or glycopeptides or bacitracin or A22.

We appreciate the remark. We have carried out an additional control as suggested by the reviewer by growing the cells with sublethal amounts of fosfomycin to apply PG stress while keeping growth rate constant. This stress has an effect – such as bulging in wildtype cells. We have included an image (Figure 1—figure supplement 1F) that shows that the flotillin mutant also suffers from the fosfomycin stress. Importantly, if the flotillin deletion would lead to general PG stress such as caused by fosfomycin, the fosfomycin treated wildtype cells should look like the flotillin deletion without fosfomycin – which is not the case.

We included a remark to state that flotillin deletion does cause peptidoglycan synthesis stress. Other stressors were not included as they would impact on membrane processes (bacitracin, glycopeptides) or cause a potential accumulation of precursor (betalactams).

If PBP biosynthetic activity is reduced (or PG degradation increased) in the floAT mutants, would this not lead to a (slight) accumulation of lipid II at the septum as well, and this could explain the observation in Figure 1? Also, this effect on PG stress could easily explain the reduction in MreB movement.

It is true that the reduction of the activity of some specific PBPs (2a and H) can lead to reduced MreB movement, but it is not evident that this of itself would result in increased LipidII at the septum – one could also argue that the unprocessed LipidII accumulates at the lateral wall. But a general reduction in biosynthetic activity or increase of degradation would also be expected to affect septum formation.

Finally, my understanding was that an increase in lipidII would INCREASE membrane fluidity, but they show in Figure 3 that fluidity is reduced in the floAT mutant.

We have tried to explain this point better in the text – the overall fluidity of the membrane is reduced, but a shift resulting in a relative local increase of disordered lipids and LipidII at the septum would still keep the membrane more fluid at the septum relative to the rest of the cell.

2) Growth rate effect: I disagree that the difference between LB and SMM can be pinned on growth rate. The cellular metabolism is completely different in minimal vs. rich medium so how can the authors rule out that a change in biosynthetic pathways does not affect lipid composition, PG (bactoprenol or lipidII) synthesis, isoprenoids or simply certain membrane proteins, all of which could alter membrane fluidity. The growth rate argument does not hold here, and it could only be proven if the authors stick to LB and lower growth rate there, ideally by reducing temp, but there could be complications on fluidity arising from a temperature shift. Inducing ppGpp would also be possibility, but this could also reprogram cell metabolism and make the interpretation ambiguous. Perhaps use diluted LB?

We agree with the reviewer that there are alternative explanations for the differences observed between LB and SMM. We compared these to growth rate because in the *B. subtilis* literature the same point is made about growth rate and MreB velocity, which our paper shows may have a different cause – but we do observe a correlation between growth rate, fluidity and MreB velocity. We have modified our remarks to make it clear that growth rate or growth conditions determine fluidity. We agree that this needs to be investigated more carefully but think that is beyond the scope of the current paper as various causes need to be investigated such as the presence of specific lipids in the membrane in certain growth conditions, or differences in crowding of the membrane by membrane proteins during various growth conditions. We have tried the suggested experiment with diluted LB (Author response image 1). This experiment revealed that we could lower growth rate by diluting LB up to 5 times (stronger dilutions led to excessive lysis) – but not to the growth rate observed with SMM (Author response image 1). Importantly, at all these conditions cell dimensions were still completely different. Cells grown on (diluted) LB are wider and longer than SMM grown cells (Author response image 1). Also, at all LB dilutions, the phenotype of the triple mutant remains – elongated cells and delocalized cell wall synthesis (Author response image 1).

3) The benzyl alcohol addition experiment is the best (and in my opinion only) argument for their claim on membrane fluidity, but how do we know that this compound directly affects the membrane, rather than acting indirectly, like SMM, changing metabolism?

The action of benzyl alcohol as a membrane fluidizer is well established in the literature. It acts by increasing membrane hydration, as has been shown in experiments on *B. subtilis* (Konopasek et al., 2000). We have clarified this in the text. But we agree that this was the sole experiment indicating a link between flotillins, membrane fluidity and cell wall synthesis (e.g. MreB dynamics). We therefore thought about a direct proof to show that flotillins really fluidize membranes. We teamed up with Dr. Birgit Habenstein in Bordeaux and used purified and reconstituted FloT for solid-state-NMR experiments (see answer below).

How quick are the changes observed after addition of BA? Are the changes in fluidity and MreB localization still observed when cells are pre-treated with protein synthesis inhibitors and /or cerulenin (FA inhibitor)? By the way, are FRAP measurements of membrane-tethered GFP consistent with a change in fluidity?

We appreciate the suggested experiments but have added a different, in vitro, experimental approach that provides an independent line of evidence that flotillin proteins are capable of fluidizing membranes. Using solid state NMR to determine lipid mobility in liposomes with and without reconstituted flotillin we show that the addition of FloT is sufficient to fluidize the liposomal membranes, to a much larger extent than observed for other membrane inserted proteins. This experiment has now been added to the paper in an additional section and Figure 5.

The bottom line is: I'm not a fan of the previous flotillin model either, but I don't think the authors provide compelling evidence in favor of their model. In fact, their experimental data does not completely exclude the previous model about flotillins facilitating PG assembly/turnover machines either.

We do hope that with the series of new data including the in vitro characterization of FloT, we have now proven the fluidizing effect of flotillins.

[Editors’ note: what follows is the authors’ response to the second round of review.]

Overall the study is solid, convincing, well presented and certainly suitable for publication in eLife. Nevertheless, some clarifications should be provided before the manuscript is accepted for publication.1) It is not clear why a reduction in MreB mobility in the membrane of floAT cells grown in LB would lead to increased vancomycin-FL and HADA staining at the septum. This relationship is poorly accounted for in the manuscript. MreB is not at the septum, so why would lower mobility of MreB cause an increase in septal lipid II? This observation is thus circumstantial. MreB and septal lipid II could be both affected because of reduced fluidity, but those events do not have to be linked:– Perhaps membranes are less fluid and if lipid II is inserted at the septum, it is less likely to diffuse away in floAT cells because of the reduced fluidity?– Or there is just a preference for lipid II to accumulate there because the membrane is curved and fluidity at the curvature is affected in the absence of FloAT? If that's the case, what does the additional deletion of pbp1 really tell us (although it is interesting)?

We agree with some of the possible explanations provided. We have included a new paragraph in the Discussion to clarify this issue, in which some of the possibilities raised are included, and in which the relation between MreB and septal staining is further clarified and supported with additional references (Discussion, second paragraph).

2) To clarify these issues, these questions should be discussed and it could be easier for the reader if the order of the results were turned around and tell the story more linearly. For example, rather than going from PG studies to membrane fluidity studies, the manuscript could present the established part first (NMR, fluidity, MreB) and then argue that reduced MreB mobility prompted imaging with HADA, Vanco-FL and finally check the pbp1floAT triple mutant. The current read takes a circular path (starting with PG and ending with MreB) and again there is no compelling causality between MreB movement and septal PG presented here.

We have discussed a reorganization of the text with Dr. Mignot – we found that this would cause other problems with the organization of the text and Dr. Mignot agreed that changing the organization was not a must.